# Antioxidant Activity, Metal Chelating Ability and DNA Protective Effect of the Hydroethanolic Extracts of *Crocus sativus* Stigmas, Tepals and Leaves

**DOI:** 10.3390/antiox11050932

**Published:** 2022-05-09

**Authors:** Sabir Ouahhoud, Amine Khoulati, Salma Kadda, Noureddine Bencheikh, Samira Mamri, Anas Ziani, Sanae Baddaoui, Fatima-Ezzahra Eddabbeh, Iliass Lahmass, Redouane Benabbes, Mohamed Addi, Christophe Hano, Abdeslam Asehraou, Ennouamane Saalaoui

**Affiliations:** 1Laboratory of Bioresources, Biotechnology, Ethnopharmacology and Health, Faculty of Sciences, Université Mohamed Premier, Oujda 60000, Morocco; khoulati_amine1718@ump.ac.ma (A.K.); bencheikh_noureddine1718@ump.ac.ma (N.B.); s.mamri@ump.ac.ma (S.M.); anas.ziani@ump.ac.ma (A.Z.); sanae.baddaoui@ump.ac.ma (S.B.); r.benabbes@ump.ac.ma (R.B.); a.asehraou@ump.ac.ma (A.A.); e.saalaoui@ump.ac.ma (E.S.); 2Laboratory of Improvement of Agricultural Production, Biotechnology and Environment, Department of Biology, Faculty of Sciences, Université Mohamed Premier, Oujda 60000, Morocco; s.kadda@ump.ac.ma (S.K.); m.addi@ump.ac.ma (M.A.); 3Laboratory of Bioresources, Biotechnology and Bioinformatics, Higher School of Technology of Khenifra, Université Sultane Moulay Slimane, Khénifra 54000, Morocco; fatima-ezzahra.eddabbed@usms.ma; 4Laboratory of Biotechnology, Environment, Agri-Food and Health, Faculty of Science Dhar Mahraz, Sidi Mohamed Ben Abdallah University, Fez 1796, Morocco; i.lahmass@ump.ac.ma; 5Laboratoire de Biologie des Ligneux et des Grandes Cultures, INRA USC1328, Orleans University, CEDEX 2, 45067 Orléans, France

**Keywords:** antioxidant activity, metal chelating ability, DNA protective effect, *Crocus sativus*, saffron

## Abstract

The present study investigated the antioxidant activity, metal chelating ability and genoprotective effect of the hydroethanolic extracts of *Crocus sativus* stigmas (STG), tepals (TPL) and leaves (LV). We evaluated the antioxidant and metal (Fe^2+^ and Cu^2+^) chelating activities of the stigmas, tepals and leaves of *C. sativus*. Similarly, we examined the genotoxic and DNA protective effect of these parts on rat leukocytes by comet assay. The results showed that TPL contains the best polyphenol content (64.66 µg GA eq/mg extract). The highest radical scavenging activity is shown by the TPL (DPPH radical scavenging activity: IC_50_ = 80.73 µg/mL). The same extracts gave a better ferric reducing power at a dose of 50 µg/mL, and better protective activity against β-carotene degradation (39.31% of oxidized β-carotene at a 100 µg/mL dose). In addition, they showed a good chelating ability of Fe^2+^ (48.7% at a 500 µg/mL dose) and Cu^2+^ (85.02% at a dose of 500 µg/mL). Thus, the antioxidant activity and metal chelating ability in the *C. sativus* plant is important, and it varies according to the part and dose used. In addition, pretreatment with STG, TPL and LV significantly (*p* < 0.001) protected rat leukocytes against the elevation of percent DNA in the tail, tail length and tail moment in streptozotocin- and alloxan-induced DNA damage. These results suggest that *C. sativus* by-products contain natural antioxidant, metal chelating and DNA protective compounds, which are capable of reducing the risk of cancer and other diseases associated with daily exposure to genotoxic xenobiotics.

## 1. Introduction

Saffron is a spice that has been used for more than 3000 years, and it belongs to the family Iridaceae and the genus *Crocus*, which includes about 80 species distributed mainly in the Mediterranean and in Southwest Asia. Among these species is saffron, which is the most expensive food product in the world [1,2]. Saffron is commonly cultivated in Iran, Morocco, India, Greece, Azerbaijan and Spain. Saffron is listed among Morocco’s terroir products. Thanks to the “Green Morocco” plan, the area reserved for saffron cultivation has increased from 622 ha in 2008 to 1865 hectares in 2019 [3]. With its unique bitter taste, slightly metallic notes, and hay-like aroma, saffron has found many valuable uses, from perfumes to dyes to medicines, but it is especially valued as a flavoring and coloring agent in foods [4]. Since ancient times, antidepressant, antinociceptive, anti-inflammatory, antispasmodic, stomachic and stimulant powers have been assigned to this spice [5,6]. More recent studies have proven that it has antioxidant [7], antitumor [8], anti-carcinogenic [9], antidiabetic [10,11], hepatoprotective [12,13], and nephroprotective properties [14]. Saffron production generates large amounts of by-products, which are often disposed of as unnecessary bioresidues [15,16,17]. For example, approximately 63 kg of flowers or 53 kg of tepals, 1500 kg of leaves, 100 kg of spathes and hundreds of bulbs that are too small and/or have physical or biological alterations are discarded to obtain only 1 kg of dry stigmas [18]. However, this biomass is a non-negligible source of bioactive compounds whose exploitation would greatly increase the profitability and sustainability of saffron production [19].

The three main components of saffron are crocetin esters, picrocrocin and safranal. They are mainly responsible for color, flavor and aroma, respectively [4]. Several flavonoids have been identified in saffron stigmas. Flavonols are the most abundant, and, specifically, the most abundant kaempferol derivatives [20]. The tepals of *C. sativus* are rich in bioactive compounds such as flavonols and anthocyanins. The main flavonols identified in the hydroalcoholic extract of tepals are kaempferol, quercetin and isorhamnetin glucosides. Several anthocyanins are reported, such as delphinidin, petunidin and malvidin glucosides [21,22]. The bioactive compounds of leaves are very little studied compared to other by-products of *C. sativus*. Different phenolic compounds have been identified, and kaempferol and luteolin glycosides are the major polyphenols in the leaves of *C.*
*sativus* [23,24,25].

Crocins and crocetin have low stability, low absorption and low bioavailability [26]. Studies have suggested that crocines are not absorbed after oral intake, but are significantly hydrolyzed to crocetin in the intestinal tract [27]. In addition, crocins and picrocrocin are bioaccessible (50% and 70%, respectively) under in vitro gastrointestinal digestion conditions [28]. Phenolic compounds can exist as free aglycones and as glycosides. The majority of in vivo studies show good gastric absorption of aglycones such as quercetin and daidzein, while glycosides are poorly absorbed. Aglycone isoflavones are absorbed in the stomach, while glycoside isoflavones are absorbed in the intestine [29]. Glycosylated anthocyanins are present intact in the blood, which is an exception for polyphenols. Moreover, their specific mechanism of absorption at the gastric level may be mediated via gastric bilitranslocase [30]. There are few studies in which the bioavailability of different flavonols has been compared, and their results suggest that kaempferol is absorbed more efficiently than quercetin and quercetin is more extensively metabolized into other compounds [31,32,33]. Kaempferol was found to be stable over a wide range of different pH values, whereas quercetin requires an acidic pH to avoid oxidation [33]. The prevalence of many chronic diseases is trending upward. Cardiovascular disease, obesity, diabetes, urinary stones, asthma and inflammatory bowel disease (IBD) are all rising, among others [34,35,36,37]. These chronic inflammatory diseases often exist as comorbidities with common physiological manifestations, compounding patient burden and suggesting common origins [38,39,40]. While a number of genetic and environmental factors contribute to the manifestation of chronic disease, an emerging assumption postulates that dysbiosis, an imbalance in the composition and metabolic capacity of our microbiota, increases the risk of developing chronic disease [41,42,43,44]. Plant foods can enhance inflammation, obesity and microbiota in a variety of ways. The microbiota is mainly diet-dependent, and vegetables promote good bacteria, which influence antioxidant capacity, degrade toxic metabolites and control blood pressure, atherosclerosis, hunger, metabolism and immunity [45].

The antioxidant effect of saffron by-products has already been evaluated by several studies [46,47]. The present study consists in investigating and comparing, for the first time, the antioxidant activity of hydroethanolic extracts of stigmas, tepals and leaves of *C. sativus* from the Talouine region of Morocco. In addition, this work aims to evaluate the metal chelating ability, as well as the DNA protective effect, of the hydroethanolic extracts of these by-products and compare them with those of stigmas in order to propose new valorizations, in particular, in the pharmaceutical field.

## 2. Materials and Methods

### 2.1. Chemicals

All chemicals used in this study were of analytical grade and purchased from Sigma chemicals.

### 2.2. Plant Material

Stigmas, tepals and leaves of saffron (*Crocus sativus* L.) were collected from a farm in Taliouine (30°31′54″ north, 7°55′25″ west, southern Morocco). The saffron of this location was grown without any chemical treatment. The different parts of the plant were collected between October and November of 2016. The botanical identification of the plant was done by Professor Botanist Fennane Mohammed from the Scientific Institute of Rabat, Morocco. Three species of the plant were deposited in the herbarium of the University Mohammed Premier, Oujda, Morocco under voucher number HUMPOM210.

### 2.3. Preparation of Plant Material

First, the stigmas and bio-residues of *C sativus* were separated manually. The stigmas were oven dried at 37 °C for 4 h. However, the tepals and leaves were oven dried at 37 °C for 24 h, respectively. The dried plant material was then ground using an automatic grinder [11,13].

### 2.4. Preparation of Hydroethanolic Extracts

Our preliminary studies revealed that 80% ethanol seems to be the solvent that showed the best polyphenol content and noted the best biological activities [11,13]. In addition, the FDA has listed ethanol as a Generally Recognized as Safe (GRAS) substance, which means that a group of specially qualified experts has determined that ethanol is safe and preferred for use in food products [48]. The protocol used was based on that described by Ouahhoud et al. [11]. The plant material was macerated in 80/20 (*v*/*v*) ethanol/water for 24 h under agitation in the dark and at room temperature. A plant powder/solvent ratio of 2 g per 50 mL was used for extraction. After this first extraction, the solvent was filtered (0.45 µm) and the residue was collected for the second extraction. This procedure was repeated three times and the total hydroethanolic phase was dried using the rotary evaporator at 40 °C. Finally, the dry hydroethanolic extract of stigmas (STG), tepals (TPL) and leaves (LV) was stored at −20 °C. The extraction yield (expressed as g/100 g dry matter) was 64.57% for stigmas, 69.45% for tepals and 31.19% for leaves

### 2.5. Animals

Eight-week-old male *Wistar* rats (weights 150–200 g) were maintained in the experimental animal house (in the biology department, Faculty of Sciences, Oujda, Morocco) at a constant temperature between 19 and 23 °C and were subjected to a 12 h light/12 h dark cycle. The rats had free access to water and food (provided by SONABETAIL, Oujda, Morocco) [13]. The maintenance and handling of the rats was in accordance with the conventional international standard guidelines and with the Helsinki declaration for use of laboratory animals, and the study was approved by the institutional review board of the Faculty of Sciences, Oujda, Morocco (01/21-LBBEH-04 and 15 March 2021).

### 2.6. Determination of Total Polyphenols

The protocol used was based on that described by Swain and Hillis [49], with some modifications, as follows: 0.25 mL of samples at 1 mg/mL were placed in a test tube with 1875 mL of distilled water, then 0.125 mL of Folin-Ciocalteu reagent was added, followed by the addition of 0.25 mL of 20% sodium carbonate solution. After 30 min of incubation at room temperature, the absorbance was measured at 725 nm against the blank. The calibration range was established using the gallic acid (GA) reference solution. The concentrations were calculated from the calibration line (absorbance = a (GA), where a is the slope and (GA) is the concentration of gallic acid). The results are expressed as mg gallic acid equivalent per mg dry extract.

### 2.7. Antioxidant Activity

#### 2.7.1. Antiradical Activity by DPPH Method

The effect on DPPH–radical scavenging was determined according to the method described by Sánchez-Moreno and co-workers [50], with slight modifications. Briefly, 50 µL of the sample or standard was mixed with 1950 µL of the freshly prepared DPPH solution (2.4 mg of DPPH was suspended in 100 mL of methanol). After 30 min of incubation in the dark, the absorbance was measured at 517 nm against the blank. Ascorbic acid was used as the standard and the percentage of inhibition was calculated using the following formula:

Percent inhibition = [[A_0_ − A_1_]/A_0_] × 100
where A_0_ is the absorbance of DPPH in the absence of the sample and A_1_ is the absorbance of the sample or standard.

The values of the inhibitory concentrations required for the trapping and reduction of 50% moles of the DPPH–free radical [IC_50_] were calculated from the graph by the exponential regressions of the plotted graphs, with percent inhibition as a function of the different concentrations of the tested samples.

#### 2.7.2. Ferric Reducing Antioxidant Power (FRAP)

The reducing power of iron was examined using the protocol described by Karagôyler et al. [51]. The different concentrations of the sample were diluted in methanol, then 1 mL of the sample was mixed with 2.5 mL of phosphate buffer (0.2 M, pH 6.6) and 2.5 mL of a 1% potassium ferricyanide K_3_Fe (CN)_6_ solution. Then, the reaction mixture was incubated at 50 °C for 20 min. After cooling, 2.5 mL of 10% trichloroacetic acid was added to stop the reaction. The tubes were then centrifuged at 3000 rpm for 30 min. Then, 2.5 mL of the supernatant was mixed with 2.5 mL of distilled water and 500 µL of the freshly prepared FeCl_3_ solution [0.1%]. Ascorbic acid was used as a reference and the absorbance was measured at 700 nm against the blank.

#### 2.7.3. Bleaching Test for β-Carotene

The β-carotene bleach test was evaluated according to the method described by Kabouche et al. [52]. Briefly, 2 mg of β-carotene was dissolved in 1 mL of chloroform and mixed with 20 mg of linoleic acid and 200 mg of tween 80. The chloroform was evaporated at 35 °C using a rotary evaporator. Then, a 100 mL volume of aerated distilled water was added to the flask and the resulting mixture was shaken vigorously. Then, 50 µL of sample was added to the tubes containing 2.45 mL of the β-carotene/linoleic acid emulsion. The absorbance was measured at 490 nm before (A_0_) and after (A_f_) incubation at 50 °C for 2 h. BHT was used as a reference. The negative control contained methanol instead of the extract. The percentage of oxidized β-carotene was calculated according to the following formula:


Percentage of oxidized β-carotene = [[A_0_ − A_f_]/A_0_] × 100


### 2.8. Metal Chelating Power

#### 2.8.1. Iron Chelation Test

The chelating power of ferrous iron was determined according to the method of Carter [53], with slight modifications. Indeed, 250 µL of the sample or reference (at different concentrations) was mixed with 1 mL of acetate buffer (0.1 M, pH 4.9) and 25 µL of FeCl_2_ (2 mM). The reaction mixture was incubated for 30 min at room temperature. Then, 0.1 mL of ferrozine (5 mM) was added. The reaction medium was incubated for 30 min at room temperature, allowing the complexation of ferrous iron to the purple-colored having an absorption maximum at 562 nm.

The negative control was similarly prepared by replacing the sample with the same volume of distilled water. The absorbance was measured at 562 nm against a blank containing distilled water. The percentage inhibition of Fe^2+^-ferrozine complex formation was calculated according to the following formula:


Percentage inhibition = [[A_0_ − A_1_]/A_0_] × 100


A_0_: absorbance of the negative control; and

A_1_: absorbance of the sample or reference.

#### 2.8.2. Copper Chelation Test

Copper chelation activity was determined according to the protocol described by Saiga et al. [54], with some modifications, as follows: 0.25 mL samples were mixed with 1 mL of sodium acetate buffer (50 mM, pH 6.0) and 25 µL of CuSO_4_ (5 mM). The reaction mixture was incubated at room temperature for 30 min. Then, 25 µL of purple pyrocatechol (PV) solution was added. After 30 min of incubation, the absorbance was measured at 632 nm. Distilled water and EDTA were used as the control and reference, respectively. The percentage inhibition of PV-Cu^2+^ complex formation was calculated as follows:


Percent inhibition = [[A_0_ − A_1_]/A_0_] × 100


A_0_: absorbance of the negative control; and

A_1_: absorbance of the sample or reference.

### 2.9. DNA Protective Effect

#### 2.9.1. Collection and Treatment of Cells

The rats were anesthetized with pentobarbital and retro orbital collection of blood was done in tubes containing heparin. Then, two milliliters of fresh blood from a male Wistar rat was added to 2 mL of PBS (137 mM NaCl; 2.7 mM KCl; 10 mM Na_2_HPO_4_; 1.76 mM KH_2_PO_4_; and pH 7.4, without Ca^2+^ or Mg^2+^). The diluted blood was thus put in contact with the extracts as follows:-Genotoxic effect

Ten microliters of blood cells were contacted with 200 µL of plant extract dissolved in PBS (STG, TPL or LV, with a final concentration of 75 µg/mL) for 15 min at 37 °C. Then, 200 µL of PBS (pH 7.4, without Ca^2+^ and Mg^2+^) was added to the medium.
-DNA-protective effect

The contact 10 µL of blood cells with 200 µL of plant extract was dissolved in PBS (STG, TPL or LV, with a final concentration of 75 µg/mL) for 15 min at 37 °C. Then, 200 µL of genotoxic agent (streptozotocin at 0.57 mM or alloxan at 13.4 mM) was added to the medium.
-Positive control

Rat blood cells (10 µL) were contacted with 200 µL of PBS (pH 7.4, without Ca^2+^ or Mg^2+^) for 15 min at 37 °C. Then, 200 µL of the genotoxic agent (streptozotocin at 0.57 mM/L or alloxan at 13.4 mM) was added to the medium.
-Negative control

Blood cells (10 µL) were contacted with 200 µL of PBS (pH 7.4, without Ca^2+^ or Mg^2+^) for 15 min at 37 °C. Then, 200 µL of PBS (pH 7.4, without Ca^2+^ and Mg^2+^) was added to the medium.

#### 2.9.2. Comet Assay

The alkaline comet assay was performed as described by Singh et al. (1988), with modifications [55]. After 5 min, the suspension was centrifuged at 4500 rpm for 10 min. Then, the supernatant was removed and the pellet containing leukocytes was dissolved in 1 mL of PBS. The washing procedure was repeated three times. After the last centrifugation, the pellet was dissolved in 200 µL of LMP agarose (low melting point, 0.5% *w*,*v* in PBS solution) and the mixture was placed on a slide previously covered with NMP agarose (normal melting point, 1.5% (*w*,*v*) in PBS solution). After 5 min, the slides were then immersed for 1 h in the lysis solution (2.5 M NaCl; 100 mM Na_2_EDTA; 20 mM Tris; 300 mM NaOH; 1% N-lauroylsarcosine sodium; 10% DMSO; and 1% Triton X-100) at 4 °C in the dark. After this lysis period, the slides were carefully rinsed with double distilled water. The slides were placed on the horizontal gel electrophoresis unit containing the freshly prepared electrophoresis solution (300 mM NaOH and 1 mM Na_2_EDTA, pH 13). The DNA was unwound for 20 min. The migration was performed for 20 min at a fixed voltage of 20 V and a current of 300 mA. The temperature of the electrophoresis solution needed to be maintained at 4 °C during the run and electrophoresis. After migration, the slides were immersed in the neutralization buffer (400 mM Trizma solution adjusted to pH 7.5 by HCl) for 5 min. This step was repeated three times.

After electrophoresis and neutralization, the comets were visualized using a silver staining method as described by Garcia et al. [56], with modifications. The slides were then immersed in the fixative solution of 15% trichloroacetic acid, 5% zinc sulfate heptahydrate and 5% glycerol for 10 min and rinsed several times with double distilled water. To stain the slides, 32 mL of solution A (5% sodium carbonate) and 68 mL of solution B (0.02% ammonium nitrate, 0.02% silver nitrate, 0.1% tungstosilicic acid and 0.05% formaldehyde, *v*/*v*) were carefully poured onto the slides placed in the slide box. This step was performed in the dark and maintained at room temperature. The slides were then processed on a plate containing the stop solution (acetic acid 1%) for 5 min. Then, the slides were rinsed with double distilled water and left to dry at room temperature.

#### 2.9.3. Microscopic Observation

The silver nitrate stained slides were observed under the microscope using the 400× objective [56]. The images were captured using a camera (CMEX 5000) and analyzed by the CaspLab software. The DNA damage could be estimated in a quantitative way using an image analyzer coupled to a processing software. In this study, we used the image analysis software CaspLab, which allowed for quantifying several parameters related to the DNA lesions [57]. Two replicates were performed per sample. Fifty cells were randomly selected per replicate.

### 2.10. Statistical Analysis

Statistical analysis of the data was done by one-way ANOVA using the statistical software GraphPad Prism 5.0. Differences between treatment groups were analyzed by Tukey’s honest significance test with significance levels of *p* < 0.05, *p* < 0.01 and *p* < 0.001.

## 3. Results

### 3.1. Total Polyphenol Content

Table 1 represents the polyphenol content of the different hydroethanolic extracts from the stigmas, tepals and leaves of *C. sativus*. The quantification of polyphenols contained in the extracts of the different parts of the plant of *C. sativus* was carried out by a colorimetric determination. The objective was to determine the content of total polyphenols since these organic molecules are responsible for the majority of biological properties and they play a role in preventing oxidative damage caused by free radicals that are the cause of various human diseases. The results shown in Figure 1 show that TPL contains the best polyphenol content (64.66 µg GA eq/mg extract), followed by LV extract (38.56 µg GA eq/mg extract) and, finally, STG extract (34.41 µg GA eq/mg extract).

### 3.2. Antioxidant Activity

#### 3.2.1. Antiradical Activity by DPPH Method

The anti-free radical activity attributed to ascorbic acid was the highest (IC_50_ = 2.5 µg/mL). The first remark is that the different extracts and fractions of tepals and leaves stigmas are not identical. Table 1 indicates that the highest DPPH radical scavenging activity is shown by the hydroethanolic extract of tepals (IC_50_ = 80.73 µg/mL), followed by the hydroethanolic extract of leaves (IC_50_= 101.5 µg/mL). However, the hydroethanolic extract of the stigmas revealed the lowest effect (IC_50_= 1554.37) (Table 2).

#### 3.2.2. Effect on the Ferric Reducing Antioxidant Power

The determination of the iron reducing activity was based on the reduction of Fe^3+^ to Fe^2+^ by electron transfer from an antioxidant. A high absorbance indicates a strong capacity of the extracts to act as antioxidants. The highest power of reduction of iron was attributed to ascorbic acid (0.51 for the 2 µg/mL and 1.05 for the 4 µg/mL). A comparison of the hydroethanolic extracts showed that TPL seemed to be the extract that reduced iron more (0.33 for the 25 µg/mL dose and 0.52 for the 50 µg/mL dose), followed by STG (0.28 for the 25 µg/mL dose and 0.49 for the 50 µg/mL dose), then LV (0.26 for the 25 µg/mL dose and 0.39 for the 50 µg/mL dose). The data also showed that all extracts acted in a dose-dependent manner (Figure 1).

#### 3.2.3. Effect on the Bleaching of β-Carotene

The principle of this method was based on the disappearance of the yellowish color of the β-carotene solution due to the breakage of the conjugated double bonds by the addition of the radical species generated by the autooxidation of linoleic acid and induced by heating. When the percentage of oxidized β-carotene was low, the antioxidant activity was high. The percentage of oxidized β-carotene in the negative control (without extract or BHT) was 82%. BHT revealed the most potent activity, as it marked only 19.8% of the degraded β-carotene at a 50 µg/mL dose and 13.2% of the degraded β-carotene at a 100 µg/mL dose. The hydroethanolic extracts of different parts of *C. sativus* obeyed the following order: first, the TPL extract (60.02% of the degraded β-carotene at a 50 µg/mL dose and 39.31% of oxidized β-carotene at a 100 µg/mL dose); second, the LV extract (64.17% of the degraded β-carotene at a 50 µg/mL dose and 43.92% of oxidized β-carotene at am100 µg/mL dose); and, finally, the STG extract (76.6% of the degraded β-carotene at a 50 µg/mL dose and 59.57%% of oxidized β-carotene at a 100 µg/mL dose). It was noted that all the samples studied acted in a dose-dependent manner (Figure 2).

### 3.3. Metal Chelating Power

#### 3.3.1. Iron Chelating Power

Figure 3 reveals the iron chelating power of the different hydroethanolic extracts of the stigmas, tepals and leaves of *C. sativus*. The test was used to determine if the chelating activity of Fe^2+^ is based on the chelation of this metal ion with ferrozine to produce a red colored complex. In the presence of other chelating agents, the formation of the Fe^2+^-ferrozine complex was disrupted and the red color of the complexes decreased. The measurement of the percentage of inhibition therefore allowed us to estimate the chelating power of iron. EDTA recorded a 31.4% iron chelation percentage at a 20 µg/mL dose and 60.2% at a 40 µg/mL dose. Comparing the hydroethanolic extracts of the three parts studied, it can be seen that the TPL fraction had the best antioxidant activity (31.73% at a 250 µg/mL dose and 48.7% at a 500 µg/mL dose), followed by the STG extract (25.15% at a 250 µg/mL dose and 44.96% at a 500 µg/mL dose) and, finally, the LV extract (18.88% at a 250 µg/mL dose and 33.88% at a 500 µg/mL dose). It can also be observed that the increase in activity was always proportional to the dose used in all extracts and all fractions (Figure 3).

#### 3.3.2. Copper Chelating Power

Figure 4 shows the copper chelating power of the different hydroethanolic extracts of the stigmas, tepals and leaves of *C. sativus*. The determination of Cu^2+^ chelating activity was based on the formation of the blue Cu^2+^-PV complex. The blue color turns to yellow in the presence of chelating agents that compete with PV. The chelating activity can therefore be estimated by measuring the rate of color reduction. EDTA marked a percentage of copper chelation of 71.9 at a 40 µg/mL concentration and 81.06% at a 60 µg/mL concentration. The hydroethanolic extracts of the different parts of *C. sativus* obeyed the following order: first, the TPL extract (68.4% at a 250 µg/mL dose and 85.02% at a dose of 500 µg/mL); second, the LV extract (49.71% at a 250 µg/mL dose and 71.50% at a 500 µg/mL dose); and, finally, the STG extract (22.64% at a 250 µg/mL dose and 38.66% at a 500 µg/mL dose). All the samples studied acted in a dose-dependent manner (Figure 4).

### 3.4. Genotoxic Effect on Rat Leukocytes

Figure 5 represents the effect of STG, TPL and LV on the percentage of DNA in the tail (A), tail length (B) and tail moment (C). The treatment of blood cells with STG, TPL and LV did not reveal a significant effect on the percentage of DNA in the tail compared with the PBS (control) group. In addition, the observed decrease in tail length and tail moment was not significant compared with the PBS group.

### 3.5. DNA-Protective Effect in Streptozotocin-Intoxicated Leukocytes

Figure 6A–C represents the protective effect of STG, TPL and LV against the increase of DNA percentage in tail (A), tail length (B) and tail moment (C) in rat leukocytes treated with streptozotocin (0.57 mM). The results obtained show that treatment with streptozotocin (0.57 mM) caused a significant increase (*p* < 0.001) in percent tail DNA, tail length and tail moment compared to the PBS treated cells. However, pretreatment with stigma, tepal and leaf extracts significantly (*p* < 0.001) protected against the elevation of percent DNA in the tail, tail length and tail moment in streptozotocin-intoxicated cells.

### 3.6. DNA-Protective Effect in Alloxan-Intoxicated Leukocytes

Figure 7A–C represents the protective effect of STG, TPL and LV against the increase of DNA percentage in tail (A), tail length (B) and tail moment (C) in rat leukocytes treated with alloxan (13.4 mM). Treatment with alloxan (ALX) (13.4 mM) induced a significant increase (*p* < 0.001) in percent tail DNA, tail length and tail moment compared to the PBS (control) treated cells. However, the incubation of leukocytes with extracts of stigmas, tepals and leaves significantly (*p* < 0.001) protected against alloxan-induced elevation of DNA percentage in the tail, tail length and tail moment.

STG refers to the hydroethanolic extract of stigmas, TPL to the hydroethanolic extract of tepals, LV to the hydroethanolic extract of leaves, PBS to phosphate buffer saline and ALX to alloxan.

## 4. Discussion

The results obtained by the determination of polyphenols by the Folin–Ciocalteu method showed that all the extracts studied contained varying amounts of these bioactive molecules. Wali et al. reported that the total polyphenol content in the ethanolic extract of *Crocus sativus* tepals was 83.98 µg GA eq/mg extract [46]. This value was higher than that of our hydroethanolic extract (64.66 µg GA eq/mg extract). The polyphenol content in *C. sativus* by-products was also examined by Lahmass et al. [47], who obtained lower values than our products. The same study revealed 23.32 µg GA eq/mg ethanolic extract of leaves and 16.63 µg GA eq/mg ethanolic extract of stigmas [47].

The three main components of saffron are crocetin esters, picrocrocin and safranal. They are mainly responsible for color, flavor and aroma, respectively [4]. Crocines constitute the family of C20-carotenoids, with both ends being esterified by sugars of glycosyl, gentiobiosyl, 3-3-D-glucosyl or neapolitanosyl nature. The extremities can have the same radicals or two different radicals [58]. The majority compound is α-crocine or ester-di-(D-gentiobiosyl)-crocetine. It is the most abundant crocin, and it can constitute more than 10% of the dry matter of stigmas [59]. Several flavonoids have been identified in saffron stigmas. Flavonols in saffron are abundant, and specifically, kaempferol derivatives such as kaempferol-3-sophoroside, kaempferol-3-sophoroside-7-glucoside, kaempferol-3,7,4′-O-triglucoside, kaempferol tetrahexoside and kaempferol-3-dihexoside [20] are the most abundant in saffron.

The tepals of *C. sativus* are rich in bioactive compounds such as flavonols and anthocyanins. The main flavonols identified in the hydroalcoholic extract of tepals are kaempferol, quercetin and isorhamnetin glucosides. Several anthocyanins are reported such as delphinidin, petunidin and malvidin glucosides [21]. The kaempferols, in decreasing order according to their concentration, are as follows: kaempferol 3-O-sophoroside, kaempferol aglycone, kaempferol 3-O-sophoroside-7-O-glucoside, kaempferol 3-O-rutinoside, kaempferol 3,7-di-O-glucoside, kaempferol 7-Oglucoside, kaempferol 3,7,4′-tri-O-glucoside and kaempferol 3-O-glucoside [29]. The amount of this flavonol in saffron tepals, in g/kg, is about 100 times higher than in other foods considered “rich” in kaempferol [60]. The main flavonol identified is Kaempferol 3-O-sophoroside. Kaempferol glycosides are the majority compounds in *C sativus* tepals (accounting for 80% of the total flavonol content). Delphinidin 3,7-O-diglucoside is the main anthocyanin identified [21,22].

The bioactive compounds of leaves are very little studied compared to other by-products of *C. sativus*. Different phenolic compounds have been identified. Kaempferol and luteolin glycosides are the major polyphenols in the leaves of *C. sativus* [23,24,25].

In vitro studies have shown that saffron crocines are probably not bioavailable in the systemic compartment after oral administration because they are rapidly hydrolysed, notably by the enzymes of the intestinal epithelium, and, to a lesser extent, by the intestinal microbiota, into trans-crocetine, which is absorbed by passive diffusion through the intestinal barrier [61]. When crocin is administered orally, it is hydrolyzed into trans- and cis-crocretin, and thus passes into the bloodstream. Subsequently, trans-crocetin can be partially conjugated with mono- and di-glucuronides in the intestinal lumen, intestinal mucosa and liver, and experience enterohepatic circulation [62]. The pharmacokinetics of crocetin have also been shown to be dose-dependent [63]. The benefits of phenolic compounds are dependent on their absorption and propagation in tissues and cells. There are significant variations in the pharmacokinetics of different types of flavonoids. Absorption rates in the small intestine range from 0 to 60% of the average dose, and elimination half-life varies between 2 and 28 h [64]. The majority of in vivo studies show good gastric absorption of aglycones such as quercetin and daidzein, while glycosides are poorly absorbed. Aglycone isoflavones are absorbed in the stomach, while glycoside isoflavones are absorbed in the intestine [26]. Glycosylated anthocyanins are present, intact, in the blood, which is an exception for polyphenols. Moreover, their specific mechanism of absorption at the gastric level may be mediated via gastric bilitranslocase [30]. The pharmacokinetics of kaempferol were examined in rats after oral and intravenous administration, confirming an absolute bioavailability of kaempferol of 11.0% [18]. Kaempferol has a low water solubility, and, therefore, a low oral bioavailability [65].

The gut microbiota catabolizes phenolic compounds into smaller molecules that are better absorbed and may have biological health benefits in the gut, as well as circulating in the plasma. In turn, phenolics compounds modulate the microbiota by promoting a “prebiotic-type effect” with the growth of beneficial microorganisms such as *Akkermansia* spp. and *Faecalibacterium* spp. and by decreasing the ratio of Firmicutes/Bacteriodetes, which is considered a beneficial effect [66]. Population differences in the intestinal microbiota determine the metabolism of phenolic compounds, producing different metabolites, that is, different metabotypes may modulate different health effects. In addition, some metabolites are more abundant in diseases or disorders related to dysbiosis of the gut flora [67]. Dysbiosis can be described as aberrant microflora which probably have an effect on immune function, as depletion of commensal species is related to the depletion of immune cell populations crucial for the coordination of immune response [68]. Concomitant diseases with microbial dysbiosis include auto-immune and allergic diseases [69,70], IBD, obesity [71], diabetes, metabolic syndrome [72] and colorectal cancer [67]. Phenolic compounds may have an effect on the composition of the gut microbiota [73,74,75,76]. These compounds significantly stimulate the growth of Enterococcus spp, Bifidobac Terium spp. and Lactobacillus [74], suggesting that phenolic compounds positively select beneficial participants of the gut microbial community [74] and modulating the gut microbiota, which contributes to improved human health, such as cardio-protective effects, nephro-protective effect, antitoxic capacity, anti-inflammatory properties, reduction of diabetes risk and inhibition of tumor cell growth, especially colonic tumor cells [45,74,77,78,79,80,81,82].

The DPPH method is a widely used method in the determination of antioxidant activity of pure compounds, as well as different extracts of aromatic and medicinal plants [83,84]. Several studies have reported the anti-radical effect of *C. sativus* by-products. Indeed, Wali et al. reported that the ethanolic extract of Kashmiri tepals indicates a significant antiradical activity, with IC_50_ = 86.6 µg/mL [46]. This value is close to that obtained by the hydroethanolic extract of tepals whose IC_50_ value is 80.73 µg/mL. Moreover, Mir et al. [85] studied the antiradical effect of different parts of *C. sativus* from Kashmir; among these parts were tepals and leaves. The authors reported that the IC_50_ values in the methanolic extract of tepals and leaves are 93.6 µg/mL and 98.8 µg/mL, respectively [85]. Lahmass et al. examined the antioxidant effect of the few by-products of *C. sativus* by DPPH assay. Indeed, the free radical scavenging activity of leaf ethanolic extract (IC_50_ = 488 µg/mL) was higher than that of stigma ethanolic extract (IC_50_ = 1533 µg/mL) [47]. In this work, the IC_50_ values reported by the hydroethanolic extracts of leaves and stigmas are 101.5 µg/mL and 1554.37 µg/mL, respectively.

It has been reported that the reducing power of some plant species is probably due to the presence of hydroxyl groups in phenolic compounds that can serve as electron donors. Therefore, antioxidants are considered as reducing and inactivating oxidants [86]. Some previous studies have also shown that the reducing power of a compound can serve as a significant indicator of its potential antioxidant activity [87,88]. Our results indicate that the iron reducing power of our products varies depending on the part, dose and extract used. Similarly, the research of Lahmass et al. reported that *C. sativus* by-products possess significant iron reducing power. Indeed, the reducing power of ethaolic extracts from stigmas was higher than those from leaves [47]. This is in agreement with our result, where we found the same order. In another study, the hydroethanol extract of tepals showed a lower Fe^3+^ reduction than the hydroethanol extract of leaves [89]. On the contrary, we found that the hydroethanolic extract of tepals is rather efficient compared to the hydroethanolic extract of leaves.

The addition of pure antioxidants [90] or in the form of plant extracts [91,92] causes a delay in the kinetics of β-carotene discoloration. According to Liyana-Pathriana and Shahidi, an extract that inhibits or delays β-carotene bleaching can be described as an antioxidant [93]. The hydroethanol and methanol extracts of different parts of *Crocus sativus* obeyed the following order: tepals > leaves > stigmas. Further, Lahmass et al. [47] noted that the ethanolic extract of leaves had a marked power against β-carotene degradation greater than that of the ethanolic extract of stigmas [47]. In addition, Sánchez-Vioque et al. [36] studied the antioxidant effect of tepals and leaves using the β-carotene bleaching method, and they noted that the hydroethanolic extract of the leaves is more effective than the hydroethanolic extract of tepals [89]. Antioxidant properties, especially anti-free radical capacity, are very important due to the detrimental effect of free radicals from food and biological systems, exacerbating cell damage and aging [94].

The metal chelating power of saffron by-products was also evaluated by the study of Sánchez-Vioque et al. [89], where the authors reported that the leaf extract showed the best Fe^2+^ chelating activity with a maximum value of 31% at 490 µg/mL, while the tepals extract revealed negligible Fe^2+^ chelating activity. The best Cu^2+^ chelating activity was observed in the leaf extract with a maximum value of 79% at 294 µg/mL, while the tepal extract reached chelating activities of 50% at the same concentration. On the other hand, our results showed that it is the hydroethanolic extract of tepals that presents the best iron chelating activity (48.7% at a dose of 500 µg/mL), followed by the STG extract (44.96% at a dose of 500 µg/mL) and, finally, the LV extract (33.88% at a dose of 500 µg/mL). Regarding the chelating power of copper, we found that the hydroethanolic extracts of different parts of *Crocus sativus* obeyed the following order: first, the TPL extract (85.02% at a dose of 500 µg/mL); secondly, the LV extract (71.50% at a 500 µg/mL dose); and, finally, the STG extract (38.66% at a 500 µg/mL dose). In biological systems, metals such as iron, copper, chromium, cobalt and others follow redox cycle reactions and possess the ability to generate reactive radicals such as superoxide anion radicals and nitric oxide. The excessive accumulation of metal ions results in oxidative stress due to increased formation of ROS, which is responsible for DNA damage, lipid peroxidation, protein modifications and other disturbances, thus leading to many diseases such as cancer, cardiovascular diseases, diabetes, atherosclerosis, neurological disorders (Alzheimer’s and Parkinson’s disease), chronic inflammation and others [95]. Chelation of these metals prevents their participation in redox reactions, thus avoiding further oxidative damage.

Compiling all the results we have obtained, we can conclude that the variation in values revealed by various studies is probably due to the origin of the plant material, the harvesting period, the extraction conditions and the experimental protocol used. Similarly, it seems that the antioxidant activities observed in the different extracts of *C. sativus* by-products are probably due to the quantity and quality of the specific types of compounds.

The genotoxicity test is considered the first test to be performed in order to assess the health safety of a compound, drug or nutraceutical [96]. The comet assay, or Single Cell Gel Electrophoresis (SCGE), is an electrophoresis technique on agarose microgel. It is a rapid and highly sensitive microscopic method for quantifying DNA damage (e.g., single and double strand breaks, oxidative damage, and DNA–protein interactions) in eukaryotic cells, as well as in some prokaryotic cells, both in vitro and in vivo [97,98,99,100]. The alkaline version of the comet assay was specifically developed to detect single-strand breaks and alkali-labile sites [55].

Our results confirm those of several previous studies that reported that *C. sativus* stigmas and their bioactive compounds, such as crocin and dimethylcrocin, are not mutagenic or genotoxic [101,102]. Similarly, studies, carried out to evaluate the antigenotoxic effect, have found that the aqueous extract of stigmas did not present any genotoxic effect on the different organs of mice by the comet test [103,104,105].

Alloxane and streptozotocin are largely used to induce experimental diabetes in animals. The mechanism of their action on pancreatic B cells has been intensively studied and is now well understood. The cytotoxic action of these two diabetogenic agents is mediated by reactive oxygen species; however, the source of their generation is different in the case of alloxan and streptozotocin. It is well known that alloxan causes diabetes by reducing alloxan to dialuric acid through redox cycling with the formation of superoxide anions. These anions undergo dismutation to hydrogen peroxide [106,107]. Through the Fenton reaction, hydroxyl radicals can be formed if divalent metals like iron and copper are present. Under in vivo conditions, these reactive oxygen species can cause DNA damage in pancreatic β-cells [107,108]. Streptozotocin enters the B cell via a glucose transporter (GLUT2) and causes DNA alkylation [109,110]. DNA damage causes the activation of poly ADP-ribosylation [111]. Poly ADP-ribosylation leads to a reduction in cellular NAD+ and ATP [112]. Increased dephosphorylation of ATP after streptozotocin treatment produces a substrate for xanthine oxidase, resulting in the generation of superoxide radicals [112]. Consequently, hydrogen peroxide and hydroxyl radicals are also generated [112]. In addition, streptozotocin releases toxic amounts of nitric oxide, which inhibits aconitase activity and contributes to DNA damage [113,114]. As a result of the action of streptozotocin, B cells undergo destruction by necrosis [115].

The comet assay, performed on rat leukocytes, revealed that the stigmas, tepals and leaves extracts of *C. sativus* appear to be non-genotoxic at the dose used, which encouraged us to go and investigate the antigenotoxic effect. We found that streptozotocin caused DNA damage in leukocytes, a result confirmed by those of Karuna et al. [116]. They reported that STZ caused an increase in comet tail length in rat lymphocytes. Several studies have shown that it causes DNA damage in different cell varieties in animal modules [117,118]. In addition, streptozotocin has been reported to cause greater DNA damage in kidney and liver cells in rats [119,120]. The decrease in degrees of STZ-induced DNA damage in leukocytes pretreated with STG, TPL and LV (in vitro) can be attributed to the protective effect against DNA alkylation and/or to the possible direct antioxidant capacity of our extracts. The phytochemical constituents of *C. sativus* may be responsible for ROS scavenging and DNA protection against streptozotocin-induced damage.

Our result confirmed that alloxan caused DNA damage in rat leukocytes. These data are similar to those of Blasiak et al., who reported in vitro, alloxan-induced DNA damage in human lymphocytes, and that free radicals are involved in the formation of this damage [121]. Other studies, performed in vivo in mice, reported alloxan-induced DNA damage in pancreatic, liver and kidney cells [122,123]. The positive effect on alloxan-induced DNA damage in STG-, TPL- and LV-pretreated leukocytes (in vitro) is probably explained by their free radical scavenging and metal chelating abilities.

In sum, the use of natural products with antioxidant and metal chelating activities may contribute to the reduction of genotoxicity caused by reactive oxygen species, and thus delay or decrease the risk of developing many chronic diseases induced by free radicals. Indeed, it has been confirmed that stigma extract can protect against the genotoxicity of some antitumor agents, including cisplatin, cyclophosphamide, mitomycin-C and methyl methanesulfonate (MMS) in animals via the modulation of lipid peroxidation, the antioxidant system and the detoxification system [124]. Further, the studies conducted by Hosseinzadeh and colleagues proved that the treatment of mice with aqueous stigma extract, crocin and safranal reduced the genotoxic effect of MMS in the kidney, lungs, and spleen [97,103]. In addition, another study showed that crocetin decreased Paraquat (PQ)-induced DNA damage in rat hepatocytes [125]. It has been reported that tepals and leaves are considered as sources of antioxidant molecules such as flavonoids. A study conducted on kaempferol, the most abundant flavonoid in the tepals and leaves of *Crocus sativus*, indicated that this compound protects against H_2_O_2_-induced DNA damage [126]. Further, the study by Anderson and colleagues reported that some flavonoids such as kaempferol, quercetin and rutin showed an antigenotoxic effect on human lymphocytes and sperm [127]. In addition, several studies have suggested that flavonoids possess antigenotoxic effects on various kinds of cells [128,129,130].

Our study dealt with a crucial and actual subject, which aims at valorizing agricultural waste. The results obtained during the different tests have confirmed the interest of the exploitation of these by-products. It would be interesting to make formulations from the by-products of *C sativus* and other medicinal plants already evaluated in order to study and search for a synergy or potentiation effect. Moreover, it would be of great interest to confirm these activities by preclinical and clinical trials, following a properly established protocol, in order to develop powerful dietary supplements or phytomedicines.

## 5. Conclusions

To conclude, the present results strongly suggest that *C. sativus* by-products contain natural antioxidant, metal chelating and genoprotective compounds, which may be capable of reducing the risk of cancer and other diseases associated with daily exposure to genotoxic xenobiotics. The data that we have generated open several avenues of research, but also leave many questions unanswered. To resolve these questions, further studies should be considered. For our perspective, we envisage to purify and search for the active molecule(s) that act specifically and understand the mechanism(s) of action of all the observed properties.

## Figures and Tables

**Figure 1 antioxidants-11-00932-f001:**
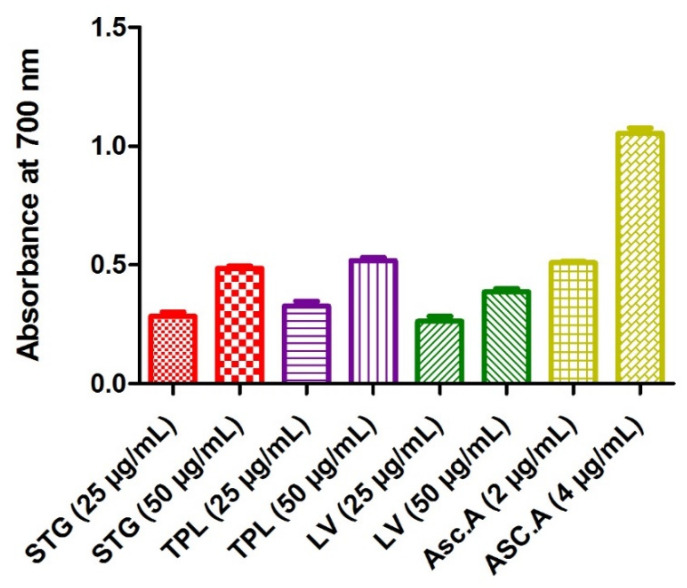
Iron reducing power of the hydroethanolic extracts of the stigmas, tepals and leaves of *C. sativus.* STG = hydroethanolic extract of the stigmas; TPL = hydroethanolic extract of the tepals; FV = hydroethanolic extract of the leaves; Asc.A = ascorbic acid.

**Figure 2 antioxidants-11-00932-f002:**
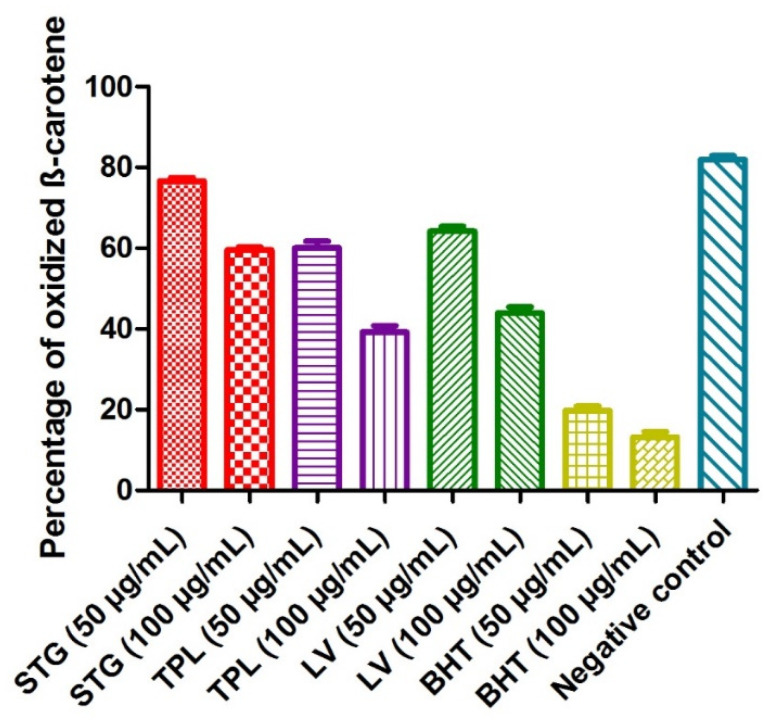
The effect of the different hydroethanolic extracts of the stigmas, tepals and leaves of *C. sativus* on β-carotene degradation. Values are expressed as mean ± SEM (*n* = 3). STG = hydroethanolic extract of the stigmas; TPL = hydroethanolic extract of the tepals; LV = hydroethanolic extract of the leaves; BHT= butylhydroxytoluene.

**Figure 3 antioxidants-11-00932-f003:**
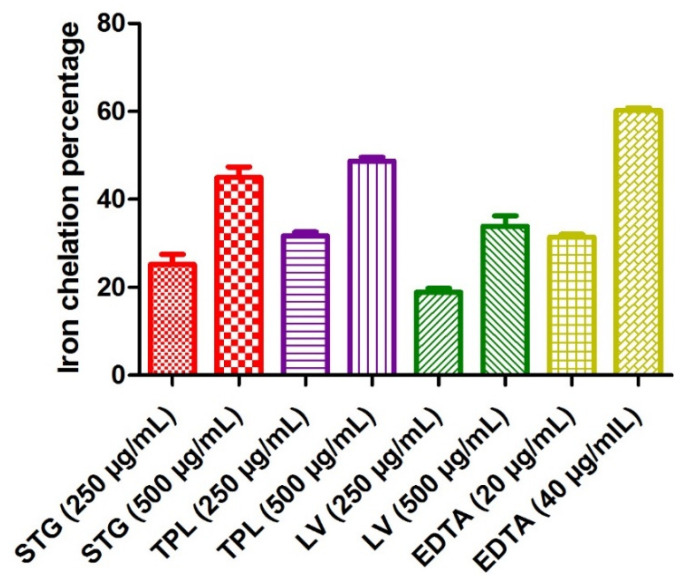
The iron chelating power of the different hydroethanolic extracts of the stigmas, tepals and leaves of *C. sativus.* Values are expressed as mean ± SEM (*n* = 3). STG = hydroethanolic extract of the stigmas; TPL = hydroethanolic extract of the tepals; LV = hydroethanolic extract of the leaves; EDTA= ethylenediaminetetraacetic acid disodium salt.

**Figure 4 antioxidants-11-00932-f004:**
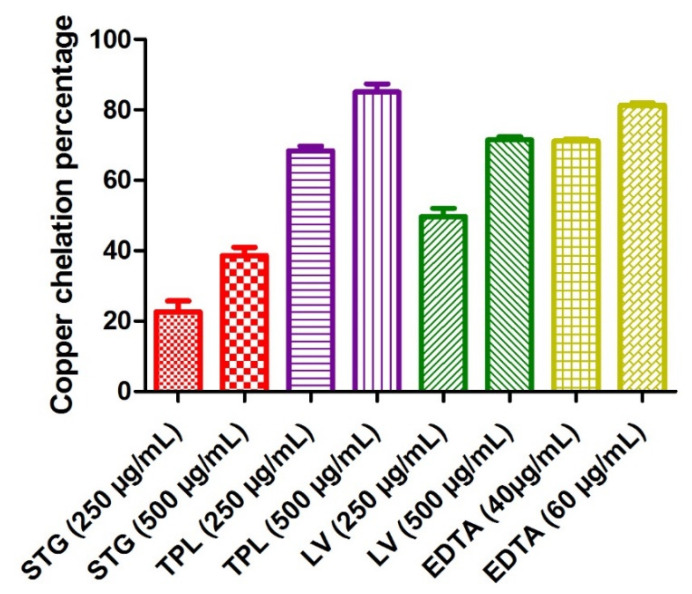
Copper chelating capacities of the different hydroethanolic extracts of the stigmas, tepals and leaves of *C. sativus.* Values are expressed as mean ± SEM (*n* = 3). STG = hydroethanolic extract of the stigmas; TPL = hydroethanolic extract of the tepals; LV = hydroethanolic extract of the leaves; EDTA= ethylenediaminetetraacetic acid disodium salt.

**Figure 5 antioxidants-11-00932-f005:**
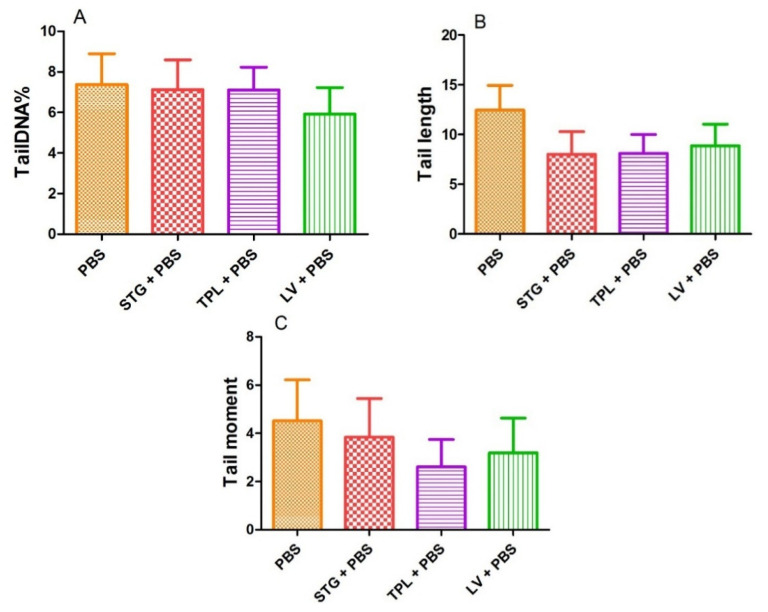
Effect of the extracts from the stigmas (75 µg/mL), tepals (75 µg/mL) and leaves (75 µg/mL) of *Crocus sativus* on the percentage of DNA in the tail (**A**), tail length (**B**) and tail moment (**C**) in rat leukocytes. Values are expressed as mean ± SEM (50 cells × 2). STG = hydroethanolic extract of stigmas; TPL = hydroethanolic extract of tepals; LV = hydroethanolic extract of leaves; PBS = phosphate buffer saline.

**Figure 6 antioxidants-11-00932-f006:**
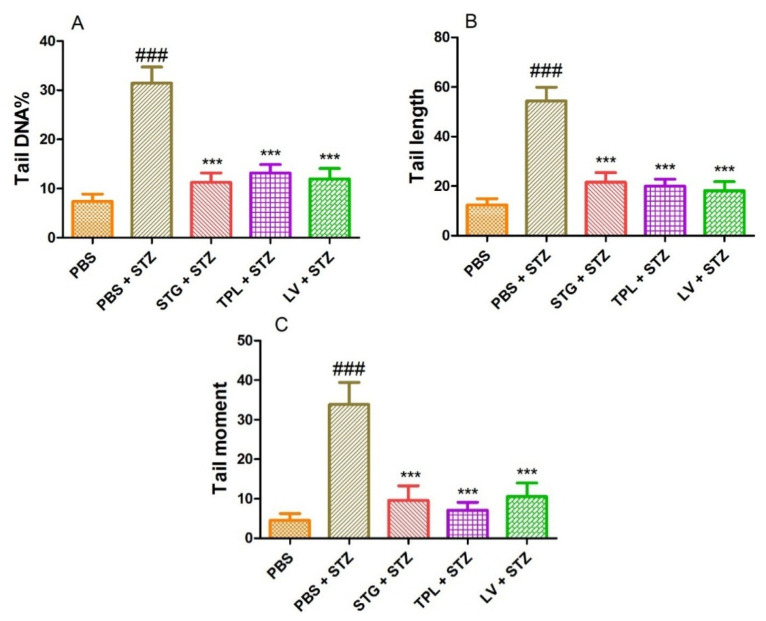
Effect of the extracts from the stigmas (75 µg/mL), tepals (75 µg/mL) and leaves (75 µg/mL) of *C. sativus* on the percentage of DNA in the tail (**A**), the length of the tail (**B**) and the tail moment (**C**) in streptozotocin-intoxicated leukocytes. Values are expressed as mean ± SEM (50 cells × 2). ### *p* < 0.001 comparison with PBS group; *** *p* < 0.001 comparison with PBS plus STZ group; STG = hydroethanolic extract of stigmas; TPL = hydroethanolic extract of tepals; LV = hydroethanolic extract of leaves; PBS = phosphate buffer saline; STZ = streptozotocin.

**Figure 7 antioxidants-11-00932-f007:**
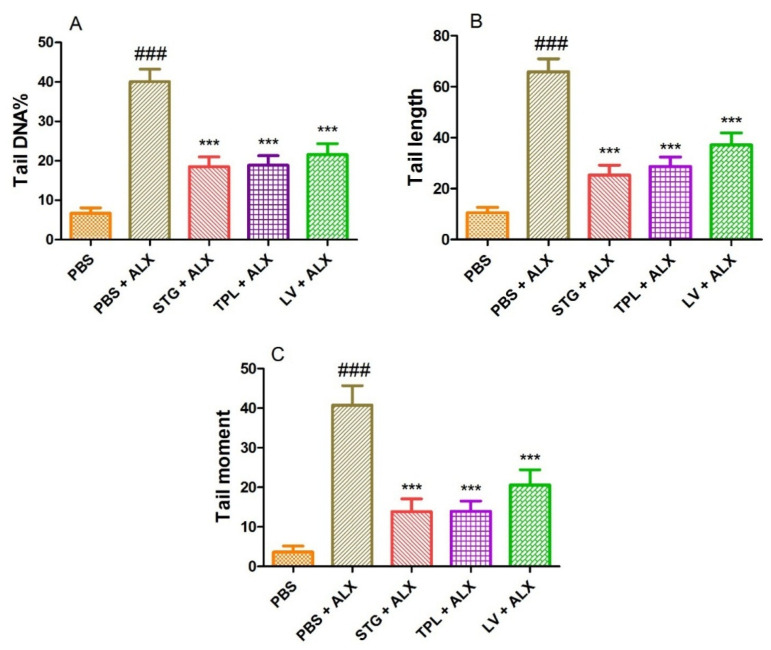
Effect of the extracts from the stigmas (75 µg/mL), tepals (75 µg/mL) and leaves (75 µg/mL) of *Crocus sativus* on the percentage of DNA in the tail (**A**), tail length (**B**) and tail moment (**C**) in alloxan-intoxicated leukocytes (13.4 mM). Values are expressed as mean ± SEM (50 cells × 2). ### *p* < 0.001 comparison with PBS group; *** *p* < 0.001 comparison with PBS plus ALX group.

**Table 1 antioxidants-11-00932-t001:** Total polyphenol content of the different hydroethanolic extracts from the stigmas, tepals and leaves of *C. sativus*.

Sample	Polyphenol Content (µg GA eq/mg Extract)
STG	(34.41 ± 1.09)
TPL	(64.66 ± 0.20)
LV	(38.56 ± 0.34)

Values are expressed as mean ± SEM (*n* = 3). GA eq: gallic acid equivalent; STG = hydroethanolic extract of stigmas; TPL = hydroethanolic extract of tepals; LV = hydroethanolic extract of leaves.

**Table 2 antioxidants-11-00932-t002:** Anti-radical activity of the hydroethanolic extracts of the stigmas, tepals and leaves of *Crocus sativus*.

Sample	IC_50_ (µg/mL)
STG	1554.37 ± 299.09
TPL	80.73 ± 0.71
LV	101.50 ± 1.55
Asc.A	2.50 ± 0.20

Values are expressed as mean ± SEM (*n* = 3). STG = hydroethanolic extract of the stigmas; TPL = hydroethanolic extract of the tepals; LV = hydroethanolic extract of the leaves; Asc.A = ascorbic acid.

## Data Availability

Data is contained within the article.

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
