# Peer review of "Antioxidant Activity, Metal Chelating Ability and DNA Protective Effect of the Hydroethanolic Extracts of Crocus sativus Stigmas, Tepals and Leaves"

_antioxidants, 2022, doi:10.3390/antiox11050932_

Round 1

Reviewer 1 Report

Dear Authors,

After the review process, I have several comments: the introduction is too poor in detail; you should include comments about expected major compounds and their bioavailability; you should include references in all Materials and Methods sections; the discussion section are many details, but future innovative valorization is not included; you are strongly encouraged to present correlations between microbiota bioactivity and bioavailability of functional compounds; the last section should be rewritten; limitation of the study should be included. Best regards!

Author Response

Response to Reviewer 1 Comments

Dear Ms. /Mr. Reviewer 1

We would like to express our appreciation for your careful reading of this work and your insightful comments that encourage and help us to improve our article. Therefore, according to your comments, we have revised thoroughly our manuscript and the response to the comments is listed below.

Point 1: the introduction is too poor in detail; you should include comments about expected major compounds and their bioavailability.

Response 1: We have modified the introduction as per reviewer suggestion (L63-L89).

Point 2: you should include references in all Materials and Methods sections.

Response 2: We have modified the Materials and Methods sections as per reviewer suggestion.

Point 3: the discussion section are many details, but future innovative valorization is not included; you are strongly encouraged to present correlations between microbiota bioactivity and bioavailability of functional compounds.

Response 3: The discussion section has been modified according to the reviewer suggestion (L455- L475 and L610-L617).

Point 4: the last section should be rewritten; limitation of the study should be included.

Response 4: The last section has been rewritten according to the reviewer suggestion (L620- L627).

We thank so much for your valuable comments that increase the quality of our manuscript.

Reviewer 2 Report

This manuscript described the radical scavenging, antioxidative, metal chelating, DNA protective activities of 80% ethanolic extracts of Crocus sativus stigma, tepal, and leaf. However, the antioxidative activities of Crocus sativus by-products were previous reported in references 28, 29 and others. It should be cited and compare the different with this article in introduction section. The reason for choice 80% ethanol as solvent to extract the Crocus sativus stigma, tepal, and leaf is critical for activities. Thus authors should explain why the reason. The mechanism for streptozotocin and alloxan is same or not should be discuss. The HPLC fingerprint  chromatograms for different extracts and contained compounds should be compare.

The other minor points were as following:

The extraction yields of different by-products should give in section 2.4.

The age of rats should be given in section 2.5.  

In L106, [GA] or [AG] should be in consistency.

In L127, [CN]6, 6 should be subscript.

In L137, the concentration of hydrogen peroxide?

In L148, FeCl2, C should be capital.

In L175, the concentration of PBS?

In L178, dissolved on à in?

In L179, FLL? The leaves extract is LV in L88.

In Table 2, the decimal number should be in consistency.

Author Response

Response to Reviewer 2 Comments

Dear Ms. /Mr. Reviewer 2

Thank you very much for your helpful comments, and time spent in the review of the paper. We have responded to your suggestions as follows:

Point 1: the antioxidative activities of Crocus sativus by-products were previous reported in references 28, 29 and others. It should be cited and compare the different with this article in introduction section.

Response 1: We have modified the introduction as per reviewer suggestion (L90-L96).

Point 2: The reason for choice 80% ethanol as solvent to extract the Crocus sativus stigma, tepal, and leaf is critical for activities. Thus authors should explain why the reason. 

Response 2: We have explained the reason for choice 80% ethanol as solvent to extract the Crocus sativus by-products as per reviewer suggestion (L115-L119).

Point 3: The mechanism for streptozotocin and alloxan is same or not should be discuss.

Response 3: The mechanism for streptozotocin and alloxan has been discussed in introduction section as per reviewer suggestion (L555- L572).

Point 4: The HPLC fingerprint chromatograms for different extracts and contained compounds should be compare.

Response 4: We are thankful to the reviewer for this insightful comment. The HPLC is a very interesting method to determine qualitatively and quantitatively the different compounds present in our samples. As per reviewer suggestion and based on our promising results we have investigated further studies to specify the phytochemical components of these C. sativus by-products. Additionally, we conducted a literature research to determine and compare the majority compounds of different studied parts of C. sativus and the data were inserted in the discussion section (L430-L454).  

Point 5: The extraction yields of different by-products should give in section 2.4.

Response 5: We have added the extraction yields of different by-products as per reviewer suggestion (L124 - L125)

Point 6: The age of rats should be given in section 2.5.  

Response 6: The age of rats has be given in section 2.5 as per reviewer suggestion (L127 – L128). 

Point 7: In L106, [GA] or [AG] should be in consistency.

Response 7: We have modified the manuscript as per reviewer suggestion (L142)

Point 8: In L127, [CN]6, 6 should be subscript.

Response 8: We have modified the manuscript as per reviewer suggestion (L163)

Point 9: In L137, the concentration of hydrogen peroxide?

Response 9: We are sorry for this error, in this experiment we used aerated distilled water and not hydrogen peroxide. We have corrected the manuscript as follow:

“Then, a 100 ml volume of aerated distilled water is added to the flask and the resulting mixture is shaken vigorously” (L173).

Point 10: In L148, FeCl2, C should be capital.

Response 10:  We have modified the manuscript as per reviewer suggestion (L184)

Point 11: In L175, the concentration of PBS?

Response 11: We have modified the manuscript as follow:

Then, two milliliters of fresh blood from a male Wistar rat is added to 2 ml of PBS (137 mM NaCl; 2,7 mM KCl; 10 mM Na2HPO4; 1,76 mM KH2PO4; pH 7.4, without Ca2+ or Mg2+) (L211 - L212).

Point 12: In L178, dissolved on à in?

Response 12: We have modified the manuscript as per reviewer suggestion (L215)

Point 13: In L179, FLL? The leaves extract is LV in L88.

Response 13: We have modified the manuscript as per reviewer suggestion

Point 14: In Table 2, the decimal number should be in consistency.

Response 14: We have modified the table 2 as per reviewer suggestion

We thank so much for your valuable comments that increase the quality of our manuscript.

Round 2

Reviewer 1 Report

Dear authors,

Related to microbiota, the authors should expand their comments to present correlation with different chronic diseases - microbiota modulation vs. dysbiosis. Please read very carefully my first comments. Best regards.

Author Response

Response to Reviewer 1 Comments

Dear Ms. /Mr. Reviewer 1

We would like to express our appreciation for your careful reading of this work and your insightful comments that encourage and help us to improve our article. Therefore, according to your comments, we have revised thoroughly our manuscript and the response to the comments is listed below.

Comment : Related to microbiota, the authors should expand their comments to present correlation with different chronic diseases - microbiota modulation vs. dysbiosis. Please read very carefully my first comments.

Response : We are grateful for this valuable comment. We have added to the previous modifications a paragraph in the introduction section (L90-L101) and another in the discussion section as per reviewer suggestion (L488-L509).

We thank so much for your valuable comments that increase the quality of our manuscript.

Reviewer 2 Report

The manuscript has been improved according to reviewers suggestion. It can be accepted in present form.  

Author Response

Thank you very much.
